**Subject Area:**
structural biology

ubiquitin, degradation, E3 ligase, BTB domain, Kelch, Cul3

**Author for correspondence:**
Alex N. Bullock
e-mail: alex.bullock@sgc.ox.ac.uk

# Identification of a PGXPP degron motif in dishevelled and structural basis for its binding to the E3 ligase KLHL12

Zhuoyao Chen[1], Gregory A. Wasney[2], Sarah Picaud[1], Panagis Filippakopoulos[1], Masoud Vedadi[2], Vincenzo D'Angiolella[3] and Alex N. Bullock[1]

[1]Structural Genomics Consortium, University of Oxford, Old Road Campus Research Building, Roosevelt Drive, Oxford OX3 7DQ, UK
[2]Structural Genomics Consortium, University of Toronto, MaRS Centre, South Tower, 101 College Street, Toronto, M5G 1L7, Canada
[3]Department of Oncology, Cancer Research UK and Medical Research Council Institute for Radiation Oncology, University of Oxford, Oxford OX3 7DQ, UK

ZC, 0000-0003-4201-4649; SP, 0000-0002-3803-4455; PF, 0000-0002-1515-1317; MV, 0000-0002-0574-0169; VD, 0000-0001-8365-9094; ANB, 0000-0001-6757-0436

Wnt signalling is dependent on dishevelled proteins (DVL1-3), which assemble an intracellular Wnt signalosome at the plasma membrane. The levels of DVL1-3 are regulated by multiple Cullin-RING E3 ligases that mediate their ubiquitination and degradation. The BTB-Kelch protein KLHL12 was the first E3 ubiquitin ligase to be identified for DVL1-3, but the molecular mechanisms determining its substrate interactions have remained unknown. Here, we mapped the interaction of DVL1-3 to a 'PGXPP' motif that is conserved in other known partners and substrates of KLHL12, including PLEKHA4, PEF1, SEC31 and DRD4. To determine the binding mechanism, we solved a 2.4 Å crystal structure of the Kelch domain of KLHL12 in complex with a DVL1 peptide that bound with low micromolar affinity. The DVL1 substrate adopted a U-shaped turn conformation that enabled hydrophobic interactions with all six blades of the Kelch domain β-propeller. In cells, the mutation or deletion of this motif reduced the binding and ubiquitination of DVL1 and increased its stability confirming this sequence as a degron motif for KLHL12 recruitment. These results define the molecular mechanisms determining DVL regulation by KLHL12 and establish the KLHL12 Kelch domain as a new protein interaction module for a novel proline-rich motif.

## 1. Introduction

Wnt signalling regulates early development and tissue homeostasis, as well as the growth of many human cancers [1,2]. In this signalling cascade, the GPCR protein Frizzled functions as a receptor for secreted Wnt ligands, which, upon binding, stimulate intracellular responses that ultimately lead to β-catenin stabilization (canonical Wnt signalling) or β-catenin-independent effects (non-canonical Wnt signalling). Dishevelled proteins (DVL1-3) form principal components of both pathways and bind to the activated Frizzled receptor inside the cell via their central PDZ domain. The DIX domain of DVLs then mediates its self-polymerization and interaction with Axin to facilitate assembly of a Wnt signalosome [3,4]. Translocation of Axin to these signalosomes blocks β-catenin degradation allowing its accumulation in the nucleus where it binds to transcriptional activators to regulate gene expression [1,5]. In addition, the DEP domain of DVLs can interact with DAAM1 to activate a β-catenin-independent pathway [6,7].

Wnt signalling is strictly controlled by the ubiquitin-proteasome system, which targets substrate proteins for degradation through the covalent attachment of ubiquitin [1,8]. An E1 enzyme first uses ATP to activate ubiquitin for covalent transfer to an E2 enzyme. E3 ubiquitin ligases further catalyse the transfer of

ubiquitin from the E2 to a substrate lysine and confer substrate specificity. To date, four E3 ligases have been reported to regulate DVL protein degradation: the HECT-family E3 ligases ITCH and NEDD4L, and the Cullin-RING E3 ligases VHL and KLHL12. ITCH specifically targets phosphorylated DVLs for proteasomal degradation [9]. NEDD4L is itself regulated by phosphorylation in response to Wnt5α signalling and mediates K6, K27 and K29-linked poly-ubiquitination of DVL2 [10]. Metabolic stress also promotes DVL2 ubiquitination by VHL that results in its aggregation and autophagic clearance [11]. By contrast, the poly-ubiquitination of DVL1-3 by KLHL12 does not require a specific cell stimulus and appears to be the result of a direct and constitutive protein–protein interaction [12,13]. Additional inhibitory factors have instead been identified that block this interaction to promote Wnt signalling. For example, NRX binds directly to DVLs to expel KLHL12 [13], while PLEKHA4 sequesters KLHL12 within PI(4,5)P₂-rich plasma membrane clusters [14]. Antagonism between KLHL12 and the abnormal spindle-like microcephaly associated protein (ASPM) is also reported to promote superpotent cancer stem cells in hepatocellular carcinoma due to the resultant increase in DVL1 protein levels [15].

KLHL12 was the first E3 to be identified for the DVLs [12], yet the molecular mechanisms determining its substrate interactions remain unknown. KLHL12 belongs to the BTB-BACK-Kelch family of proteins, which includes E3s such as KEAP1 (KLHL19) and gigaxonin (KLHL16) [16,17]. The multiple domains in these E3s facilitate their dual functions as Cullin-RING adaptors and substrate recognition modules. Interaction with Cullin3 is mediated by the BTB domain and a '3-box' motif from the BACK domain, whereas the Kelch domain mediates substrate capture [18–20]. The RING domain-containing protein Rbx1 binds to the opposite end of the Cullin3 scaffold and facilitates the recruitment of E2-ubiquitin conjugates [21,22]. Transfer of ubiquitin from the E2 to the substrate is promoted by neddylation of the Cullin scaffold [23,24]. KLHL12 can also engage target-specific co-adaptors to ubiquitinate different substrates with distinct ubiquitin chain linkages and outcomes [25]. For example, KLHL12 can assemble with the co-adaptors PEF1 and ALG2 to mono-ubiquitinate SEC31 and promote COPII complex assembly for collagen secretion [26]. In addition, KLHL12 can target the dopamine D4 receptor for both lysine and non-lysine ubiquitination [27–29].

In the absence of any known substrate recognition motifs, the structure of the Kelch domain of KLHL12 was solved previously without a bound ligand [18]. The six Kelch repeats formed the six blades (I–VI) of a canonical β-propeller fold, each individually folded into four antiparallel β-strands (βA-βD). In the current work, we address this gap in understanding, by defining a consensus recognition motif 'PGXPP' common to both substrates and co-adaptors of KLHL12. We further determined the structural basis for the binding of this motif to KLHL12 and validated this motif as a degron for DVL1 degradation in cells.

# 2. Results

## 2.1. A 'PGGPP' motif in DVL1 is critical for KLHL12 interaction

The C-terminal region of DVLs implicated in KLHL12 interaction lacks any known domains and is predicted to be structurally disordered [12]. GST pulldowns have previously demonstrated a direct interaction between the recombinant purified proteins of KLHL12 and DVL1 [13]. We therefore used the SPOT peptide technology [30] to print an array of 20-mer peptides spanning the DVL1 C-terminal residues 465–695. To map potential recruitment degron motifs in this region, we probed the array with His₆-tagged KLHL12 Kelch domain and detected bound protein by immunoblotting with anti-His antibody (figure 1a and b). A single-peptide spanning DVL1 residues 650–669 was identified that bound to KLHL12, but not to the negative control protein KLHL7 (figure 1a and b). The specific binding of this peptide to KLHL12 was confirmed using a fluorescence polarization assay, which revealed an interaction with $K_D = 22$ µM (figure 1c).

To map the minimal DVL1 epitope required for KLHL12 interaction, we used the SPOT technology for alanine scanning mutagenesis and peptide truncation experiments based on the identified 20-mer peptide. The results from these experiments were in excellent agreement and identified DVL1 residues Pro658 to Pro662 (PGGPP) as critical for KLHL12 interaction (figure 2a and b). Mutation or truncation of these residues largely abolished binding. Deletion of Val663 and Arg664 also reduced KLHL12 binding, whereas the mutation of them did not, suggesting that the backbone atoms at these positions may suffice for interaction. Other deletions and mutations outside of this region were well tolerated.

## 2.2. Structure determination for the KLHL12-dishevelled proteins1 complex

Attempts to crystallize the Kelch domain of KLHL12 in complex with the 20-mer DVL1 peptide (650–669 a.a.) were unsuccessful. However, crystals were obtained using a 15-mer peptide spanning residues 650–664 and optimized for diffraction using micro-seeding and fine screening around the initial crystallization conditions. A crystal structure of the KLHL12-DVL1 complex was subsequently solved by molecular replacement and refined at 2.38 Å resolution with four complexes in the asymmetric unit. Data collection and refinement statistics are shown in table 1.

The structure traces the Kelch domain of KLHL12 from residues 279 to 567 (figure 3a). The six Kelch repeats are folded as twisted β-sheets arranged radially around the central axis of the β-propeller. Superposition of the four complexes in the asymmetric unit reveals nearly identical conformations across the peptide-binding interface and only minor differences in the flexible loops outside this region (figure 3b). Overall, the DVL1 peptide was traced from Gly655 to Val663 (figure 3c and d; 'GGPPGGPPV'), allowing structural analysis of the critical 'PGGPP' motif. Electron density was not resolved for other N and C-terminal residues in DVL1, suggesting a lack of contacts to stabilize these flanking positions. Gly655 could only be modelled in chain F, which also lacked electron density for Pro662-Val663. Electron density for Pro662 in chains E, G and H was not as clearly resolved as that of the preceding prolines, suggesting a flexible peptide conformation at this position.

## 2.3. Interactions stabilizing the binding of dishevelled proteins1

The bound DVL1 peptide adopts a U-shaped turn conformation that is stabilized by weak intramolecular hydrogen

royalsocietypublishing.org/journal/rsob    Open Biol. **10**: 200041

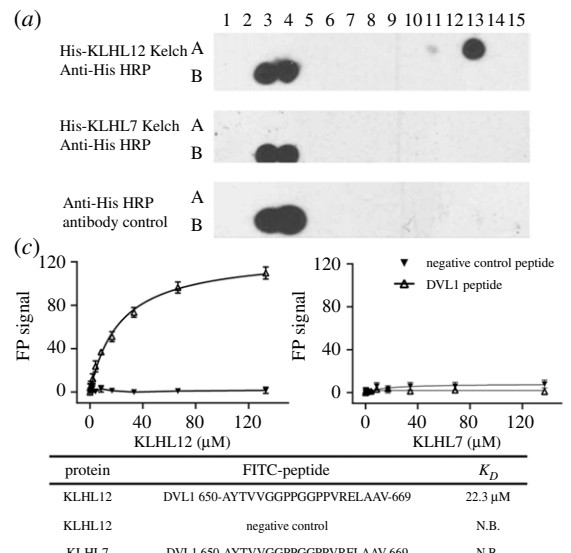

**(a)**

| | | 1 2 3 4 5 6 7 8 9 10 11 12 13 14 15 |
|---|---|---|
| His-KLHL12 Kelch Anti-His HRP | A | |
| | B | |
| His-KLHL7 Kelch Anti-His HRP | A | |
| | B | |
| Anti-His HRP antibody control | A | |
| | B | |

**(b)**

| | peptide sequences | residues | KLHL12 binding | KLHL7 binding | antibody binding |
|---|---|---|---|---|---|
| A1 | REARKYASSLLKHGFLRHTV | 465–484 | − | − | − |
| A2 | LRHTVNKITFSEQCYYVFGD | 480–499 | − | − | − |
| A3 | YVFGDLCSNLATLNLNSGSS | 495–514 | − | − | − |
| A4 | NSGSSGTSDQDTLAPLPHPA | 510–529 | − | − | − |
| A5 | LPHPAAPWPLGQGYPYQYPG | 525–544 | − | − | − |
| A6 | YQYPGPPPCFPPAYQDPGFS | 540–559 | − | − | − |
| A7 | DPGFSYGSGSTGSQQSEGSK | 555–574 | − | − | − |
| A8 | SEGSKSSGSTRSSRRAPGRE | 570–589 | − | − | − |
| A9 | APGREKERRAAGAGGSGSES | 585–604 | − | − | − |
| A10 | SGSESDHTAPSGVGSSWRER | 600–619 | − | − | − |
| A11 | PAGQLSRGSSPRSQASATAP | 620–639 | − | − | − |
| A12 | SATAPGLPPPHPTTKAYTVV | 635–654 | − | − | − |
| A13 | AYTVVGGPPGGPPVRELAAV | 650–669 | +++ | − | − |
| A14 | ELAAVPPELTGSRQSFQKAM | 665–684 | − | − | − |
| A15 | SRQSFQKAMGNPCEFFVDIM | 676–695 | − | − | − |
| B2 | DDDDDDDDDDDDDDDDDDDD | negative control | − | − | − |
| B3 | HHHHHHHHHHHHHHHHHHHH | positive control | +++ | +++ | +++ |
| B4 | HHHHHHHHHHHHHHHHHHHH | positive control | +++ | +++ | +++ |

**(c)**

Fluorescence polarization assay plots: FP signal (y-axis, 0–120) vs KLHL12 (µM) and KLHL7 (µM) (x-axis, 0–120). Legend: ▼ negative control peptide, △ DVL1 peptide.

| protein | FITC-peptide | $K_D$ |
|---|---|---|
| KLHL12 | DVL1 650-AYTVVGGPPGGPPVRELAAV-669 | 22.3 µM |
| KLHL12 | negative control | N.B. |
| KLHL7 | DVL1 650-AYTVVGGPPGGPPVRELAAV-669 | N.B. |
| KLHL7 | negative control | N.B. |

**Figure 1.** Mapping of the DVL1-binding motif for KLHL12 recruitment. (*a*) SPOT peptide array in which each spot was printed as a 20-mer DVL1 peptide with a 15 residue frameshift at each consecutive position. Spots B3 and B4 were printed as poly-His peptides as positive controls for the antibody detection. Spot B2 was printed as poly-Asp as a negative control. Arrays were incubated with purified 6 x His-KLHL12 Kelch domain, washed and then KLHL12 binding detected using anti-His HRP-conjugated antibody. As negative controls, replicate spots were probed with either antibody alone or KLHL7, a protein of the same family. Specific binding of KLHL12 was observed at spot A13. (*b*) Peptide sequences. (*c*) Fluorescence polarization assay for peptide binding to KLHL12. The KLHL12 and KLHL7 Kelch domains were assayed for binding to peptides labelled with FITC, including the spot A13 peptide (DVL1 residues 650–669) and a negative control peptide (DVL3 461-RREARKYASNLLKAGFIRHT). Spot A13 (DVL1 residues 650–669) bound to KLHL12 with $K_D = 22.3$ µM, but not to KLHL7. Data reported are from 3 replicates. The error bars show the standard error of the measurements.

**(a)**

| | DVL1 Sequences | Binding |
|---|---|---|
| A1 | H-H-H-H-H-H-H-H-H-H-H | +++ |
| B1 | | |
| C1 | A-Y-T-V-V-G-G-P-**P-G-G-P-P**-V-R-E-L-A-A-V | +++ |
| D1 | **A-A-A**-V-V-G-G-P-**P-G-G-P-P**-V-R-E-L-A-A-V | +++ |
| E1 | **A-A-A**-A-V-G-G-P-**P-G-G-P-P**-V-R-E-L-A-A-V | +++ |
| F1 | A-Y-**A-A-A**-G-G-P-**P-G-G-P-P**-V-R-E-L-A-A-V | +++ |
| G1 | A-Y-T-**A-A-A**-G-P-**P-G-G-P-P**-V-R-E-L-A-A-V | +++ |
| H1 | A-Y-T-V-V-**A-A-A**-P-**P-G-G-P-P**-V-R-E-L-A-A-V | +++ |
| I1 | A-Y-T-V-V-G-**A-A-A**-**P-G-G-P-P**-V-R-E-L-A-A-V | +++ |
| J1 | A-Y-T-V-V-G-G-**A-A-A**-G-P-P-V-R-E-L-A-A-V | − |
| K1 | A-Y-T-V-V-G-G-**A-A-A**-G-P-P-V-R-E-L-A-A-V | − |
| L1 | A-Y-T-V-V-G-G-P-**A-A-A**-P-P-V-R-E-L-A-A-V | − |
| M1 | A-Y-T-V-V-G-G-P-P-**A-A-A**-P-V-R-E-L-A-A-V | − |
| N1 | A-Y-T-V-V-G-G-P-P-G-**A-A-A**-V-R-E-L-A-A-V | − |
| O1 | A-Y-T-V-V-G-G-P-P-G-G-**A-A-A**-R-E-L-A-A-V | − |
| P1 | A-Y-T-V-V-G-G-P-P-G-G-P-**A-A-A**-E-L-A-A-V | + |
| Q1 | A-Y-T-V-V-G-G-P-**P-G-G-P-P**-**A-A-A**-L-A-A-V | +++ |
| R1 | A-Y-T-V-V-G-G-P-**P-G-G-P-P**-V-**A-A-A**-A-A-V | +++ |
| S1 | A-Y-T-V-V-G-G-P-**P-G-G-P-P**-V-R-**A-A-A**-A-V | +++ |
| T1 | A-Y-T-V-V-G-G-P-**P-G-G-P-P**-V-R-E-**A-A-A**-V | +++ |
| U1 | A-Y-T-V-V-G-G-P-**P-G-G-P-P**-V-R-E-L-**A-A-A** | +++ |

**(b)**

| | DVL1 Sequences | Binding |
|---|---|---|
| A2 | A-Y-T-V-V-G-G-P-**P-G-G-P-P**-V-R-E-L-A-A-V | +++ |
| B2 | Y-T-V-V-G-G-P-**P-G-G-P-P**-V-R-E-L-A-A-V | +++ |
| C2 | T-V-V-G-G-P-**P-G-G-P-P**-V-R-E-L-A-A-V | ++ |
| D2 | V-V-G-G-P-**P-G-G-P-P**-V-R-E-L-A-A-V | +++ |
| E2 | V-G-G-P-**P-G-G-P-P**-V-R-E-L-A-A-V | +++ |
| F2 | G-P-**P-G-G-P-P**-V-R-E-L-A-A-V | +++ |
| G2 | P-**P-G-G-P-P**-V-R-E-L-A-A-V | +++ |
| H2 | **P-G-G-P-P**-V-R-E-L-A-A-V | +++ |
| I2 | G-P-P-V-R-E-L-A-A-V | − |
| J2 | G-P-P-V-R-E-L-A-A-V | − |
| K2 | P-P-V-R-E-L-A-A-V | − |
| L2 | A-Y-T-V-V-G-G-P-**P-G-G-P-P**-V-R-E-L-A-A | +++ |
| M2 | A-Y-T-V-V-G-G-P-**P-G-G-P-P**-V-R-E-L-A | +++ |
| N2 | A-Y-T-V-V-G-G-P-**P-G-G-P-P**-V-R-E-L | +++ |
| O2 | A-Y-T-V-V-G-G-P-**P-G-G-P-P**-V-R-E | +++ |
| P2 | A-Y-T-V-V-G-G-P-**P-G-G-P-P**-V-R | +++ |
| Q2 | A-Y-T-V-V-G-G-P-**P-G-G-P-P**-V | + |
| R2 | A-Y-T-V-V-G-G-P-**P-G-G-P-P** | + |
| S2 | A-Y-T-V-V-G-G-P-**P-G-G-P** | − |
| T2 | A-Y-T-V-V-G-G-P-P-G-G | − |
| U2 | A-Y-T-V-V-G-G-P-P-G | − |

**Figure 2.** A 'PGGPP' motif in DVL1 is critical for KLHL12 interaction. DVL1 peptide variants were printed in SPOT peptide arrays exploring triple-alanine scanning mutagenesis (*a*), or N and C-terminal truncations (*b*). Arrays were incubated with purified 6 x His-KLHL12 Kelch domain, washed and then binding detected with anti-His antibody. KLHL12 binding was disrupted upon mutation or deletion of the 'PGGPP' sequence motif in DVL1. The apparent stronger binding of some truncated peptides may reflect a conformational entropic penalty for longer peptides, or differences in the efficiency of peptide synthesis or their accessibility on the membrane.

bonds between the carbonyl of Pro657 and the amides of Gly659 and Gly660 (figure 3c). This conformation allows DVL1 to form interactions with all six blades of the KLHL12 β-propeller, as shown in figure 4a and b. The majority of the interactions derive from the central 'PGGPP' motif of DVL1, in agreement with the peptide arrays. The first proline in this motif, Pro658, forms the only hydrogen bond in the complex interface through the hydroxyl of KLHL12 Tyr512 (figure 4c). Pro658 and the flanking residues Pro657 and Gly659 additionally contribute extensive hydrophobic interactions with

KLHL12 residues Tyr321, Leu371, Tyr434, Ile439 and Phe481, which span blades III, IV, V and VI (figure 4d). Further hydrophobic contacts are provided by the downstream residue Pro661, which inserts between KLHL12 Phe289, Tyr528 and Leu533 (figure 4e). Here, DVL1 Pro661 and KLHL12 Tyr528 form a prolyl-aromatic ring–ring stacking interaction. The final C-terminal proline in the 'PGGPP' motif, Pro662, is oriented away from the binding interface (figure 4e) despite the evidence of its importance in the SPOT peptide arrays (figure 2). It is likely that Pro662 fulfils a conformational role

**Table 1.** Crystallographic data collection and refinement statistics.

| structure of human KLHL12-DVL complex, PDB: 6TTK | |
| --- | --- |
| data collection | |
| beamline | diamond light source, I24 |
| wavelength (Å) | 0.9686 |
| resolution range (Å) | 79.98–2.383 (2.469–2.383) |
| space group | P 1 2$_1$ 1 |
| unit cell dimensions | |
| a,b,c (Å) | 80.225 73.145 101.845 |
| $\alpha$, $\beta$, $\gamma$ (°) | 90 94.501 90 |
| total reflections | 212 254 (31 109) |
| unique reflections | 47 094 (4664) |
| completeness (%) | 99.57 (99.59) |
| mean I/sigma(I) | 5.8 (2.2) |
| CC1/2 | 0.984 (0.816) |
| R-merge | 0.185 (0.679) |
| refinement | |
| reflections used in refinement | 47 014 (4659) |
| reflections used for R-free | 2359 (223) |
| R-work | 0.2263 (0.2889) |
| R-free | 0.2522 (0.3308) |
| number of non-hydrogen atoms | 9382 |
| RMS deviation (bonds, Å) | 0.014 |
| RMS deviation (angles,°) | 1.61 |
| Ramachandran favoured (%) | 95.90 |
| Ramachandran allowed (%) | 4.10 |
| Ramachandran outliers (%) | 0.00 |
| Rotamer outliers (%) | 0.00 |
| average B-factor (Å$^2$) | 26.11 |

[a]Values in brackets show the statistics for the highest resolution shells. RMS indicates root-mean-square.

by facilitating a slight turn in the DVL1 peptide to avoid steric hindrance with blade II of KLHL12.

## 2.4. KLHL12-induced ubiquitination and degradation of dishevelled proteins1 is dependent on the 'PGGPP' motif

To validate the identified 'PGGPP' sequence as degron motif for DVL1 recruitment and degradation by the KLHL12, we tested the binding, ubiquitination and stability of DVL1 in HEK293T cells. Two variants of full-length HA-tagged DVL1 were prepared in which the 657-'PPGGPP' motif was either mutated to 'AAAAAA' or deleted. Immunoprecipitation of the complex using full-length Flag-tagged KLHL12 was performed in cells treated with the neddylation inhibitor MLN4924 to ensure the stability of all transfected DVL1 variants. Wild-type (WT) DVL1 bound robustly to KLHL12, whereas the binding of both DVL1 mutants was abolished (figure 5a). To confirm that the interaction promoted DVL1

ubiquitination, we next transfected Flag-tagged KLHL12 and HA-DVL1 variants into cells pretreated with the proteasome inhibitor MG132. DVL1 was immunoprecipitated by anti-HA agarose and its ubiquitination status probed by immunoblotting for the presence of high molecular weight poly-ubiquitin conjugates. A marked reduction in the ubiquitination of the two DVL1 mutants was observed compared to wild-type consistent with the importance of the 'PGGPP' motif for KLHL12 recruitment (figure 5b). Some remaining background ubiquitination of the two DVL1 mutants may result from other endogenous E3 ligases, such as ITCH, NEDD4L and VHL [9–11], that probably recognize other distinct degron motifs in DVL1.

We further investigated whether the 'PGGPP' motif regulates the KLHL12-dependent degradation of DVL1. WT and mutant DVL1 constructs were therefore transfected into HEK293T cells in the presence or absence of full-length KLHL12. As shown in figure 5c, co-expression of full-length KLHL12 and DVL1 WT caused a striking reduction in the DVL1 protein level compared to expressing DVL1 WT alone. By contrast, the levels of the DVL1 mutants were unchanged by KLHL12 co-expression. The addition of MLN4924 to the cells rescued the stability of DVL1 WT confirming its dependence on Cullin-RING ligase ubiquitination activity (figure 5d).

Taken together, these data indicated that the 'PGGPP' motif was critical for both DVL1 recruitment and degradation by KLHL12.

## 2.5. 'PGXPP' is a consensus motif for interaction in other KLHL12 substrates and co-adaptors

The 'PGGPP' motif is conserved across all three DVL paralogs, except for an alanine substitution at the second glycine position in DVL2 and DVL3 (figure 6a). Identical 'PGGPP' or 'PGAPP' motifs are also present in the KLHL12 partners PLEKHA4, PEF1 and SEC31, while the dopamine D4 receptor contains a 'PGLPP' motif (figure 6a). Of note, the variant position corresponding to DVL1 Gly660 displayed the lowest buried surface area among the five residues of the 'PGGPP' motif in the DVL1 co-structure (figure 4b). To model the binding of DVL2 and DVL3, we introduced the G660A substitution into the DVL1 structure (figure 6b). The methyl side chain of this alanine was easily accommodated in the complex interface with no KLHL12 contacts within 4 Å. The binding of KLHL12 to the 'PGAPP' motif of DVL2 and DVL3 was further confirmed using a SPOT peptide array (figure 6c).

These results highlight the sequence 'PGXPP', where X represents a small non-polar residue, as a consensus site for the recruitment of interaction partners to the Kelch domain of KLHL12.

## 3. Discussion

Proline-rich domains are natively unfolded regions of high-proline content that are widely used in signal transduction processes. Short proline-rich sequence motifs within these domains facilitate the recruitment of common protein recognition domains, such as SH3, WW and EVH1 domains, which are normally associated with other effector domains [31]. DVL1-3 contain folded DIX, PDZ and DEP domains, as well as a proline-rich C-terminal domain [32]. It was found previously that KLHL12 was recruited to these C-terminal regions

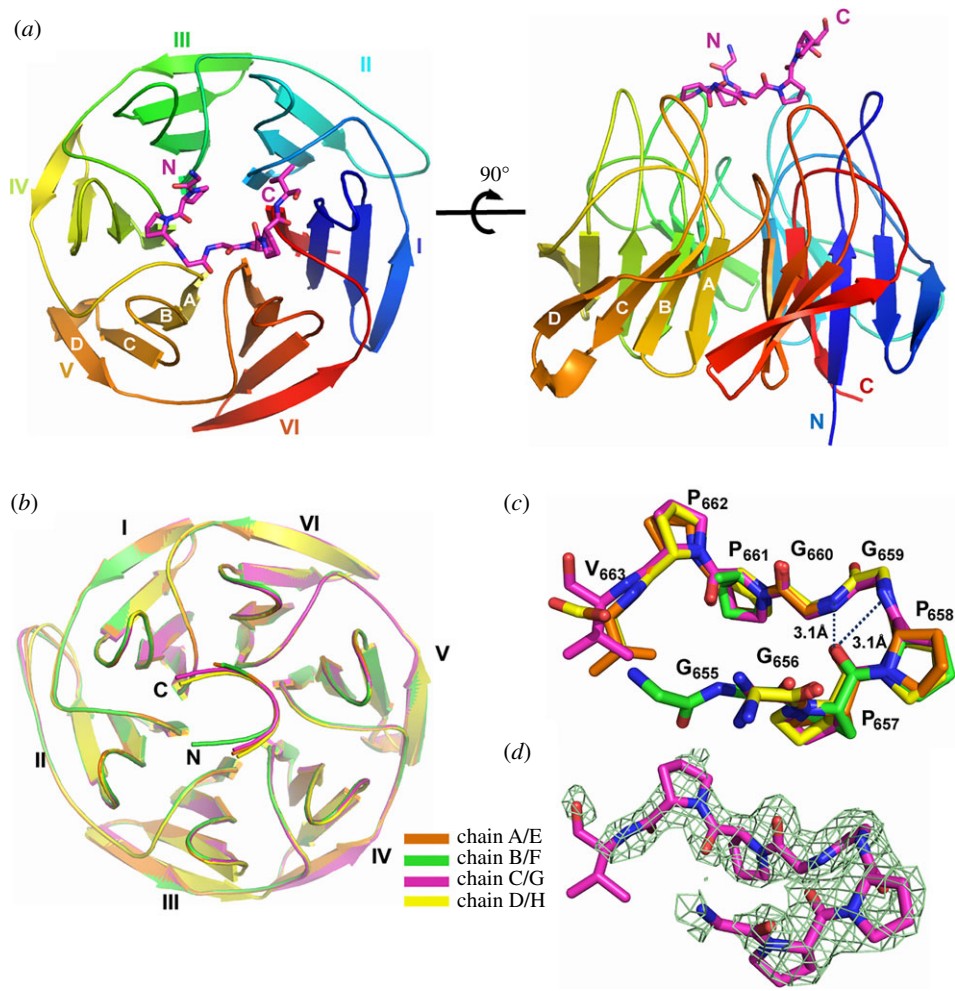

**Figure 3.** Structure of KLHL12 Kelch domain bound to DVL1 peptide. (*a*) Overview of the structure of the KLHL12 Kelch domain (rainbow ribbon) in complex with DVL1 peptide (purple sticks). Kelch repeats forming blades I to VI are labelled, as well as the four β-strands (*a*–*d*) that form each blade. (*b*) Superposition of the four KLHL12-DVL1 complexes located in the asymmetric unit. (*c*) Superposition of the four DVL1 chains in the same asymmetric unit, stick representation. Dashed lines show intramolecular hydrogen bond interactions that stabilize the DVL1 conformation together with their distances. (*d*) 2Fo-Fc electron density map (green mesh) for the DVL1 peptide contoured at 1 σ.

to mediate the ubiquitination and proteasomal degradation of DVLs [12]. In this work, we mapped this binding to a C-terminal 'PGXPP' motif that is common to DVL1-3. Mutation or deletion of this motif in DVL1 was sufficient to disrupt its binding to KLHL12 resulting in a marked reduction in DVL1 ubiquitination and increased stability. These findings confirm the importance of the 'PGXPP' sequence as a degron motif for the recruitment of KLHL12 and the subsequent degradation of DVLs. We also observed that this motif was conserved in other direct interaction partners of KLHL12, including the substrate SEC31, the co-adaptor PEF1 and the antagonist PLEKHA4.

The impact of KLHL12 on Wnt signalling was first demonstrated in zebrafish embryos in which KLHL12 overexpression was found to phenocopy DVL loss by inhibiting antero-posterior axis elongation [12]. Similar upregulation of KLHL12 activity and inhibition of Wnt signalling was observed in mammalian cells upon knockdown of the antagonist PLEKHA4 [14]. In mice, KLHL12 was found to be highly expressed in embryonic stem cells, but downregulated upon differentiation, supporting a role for KLHL12 in embryonic development [26].

The low micromolar binding determined for the DVL1 degron is comparable in affinity to the substrates of other Cullin3-dependent E3 ligases, such as KLHL20 [33] and

SPOP [19], although notably weaker than those of KEAP1 [34,35]. The interactions of KLHL12 may be further enhanced by multivalency and avidity effects. Proline-rich binders, such as DVL1-3 and PLEKHA4, are found to form high molecular weight clusters at the plasma membrane, while the KLHL12 homodimer presents two Kelch domains for their interaction. The dimerization of KLHL12 is also proposed as a mechanism to allow the Kelch domains to bind simultaneously to co-adaptors and substrates [25].

The modest affinity of the 'PGXPP' sequence is consistent with the PXXP motifs that bind with micromolar affinities to SH3 domain-containing proteins. However, their binding modes are quite distinct. Our crystal structure of the KLHL12-DVL1 complex reveals a U-shaped turn conformation for the DVL1 peptide that differs significantly from the extended conformations of the PXXP motif peptides that bind to SH3 domains. Despite the different crystal packing, the structure of KLHL12 in the DVL1 complex is unchanged compared to the unbound Kelch domain structure (0.99 Å RMSD for all atoms) [18]. Thus, DVL1 appears to bind to a preformed hydrophobic pocket in KLHL12 that is shallow and exposed. Similar to other Kelch-substrate complexes, the pocket periphery is framed by the central BC loops that protrude from each blade of the β-propeller, whereas the floor of the pocket is shaped by the DA loops that connect adjacent blades.

royalsocietypublishing.org/journal/rsob    Open Biol. **10**: 200041

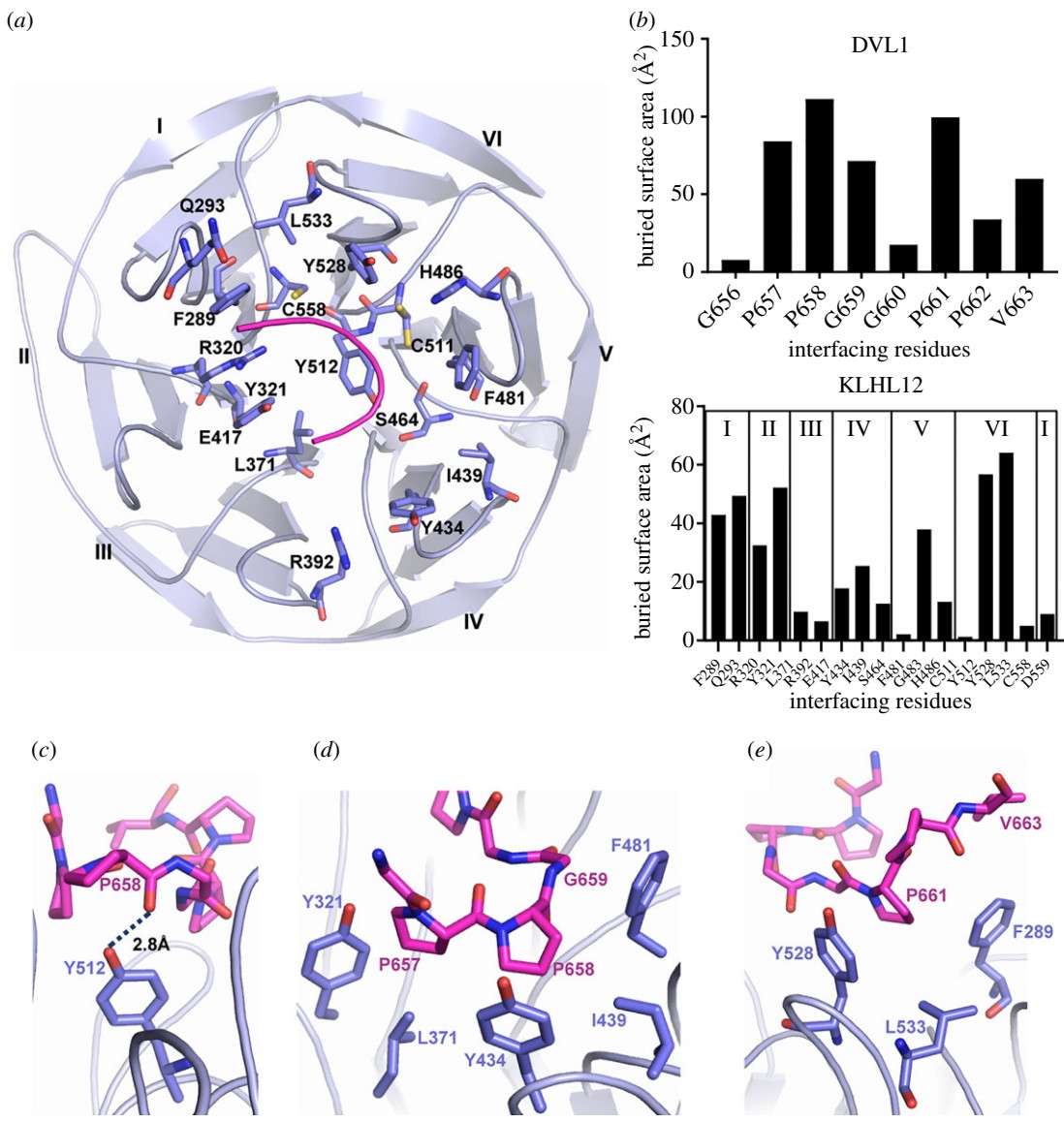

**Figure 4.** Interactions in the KLHL12 Kelch domain-DVL1 complex. (*a*) An overview of the DVL1-binding residues in KLHL12 (purple ribbon and sticks). DVL1 peptide is shown as pink ribbon. (*b*) Buried interface surface areas for interacting residues in the KLHL12 Kelch domain-DVL1 complex. (*c*) DVL1 Pro658 forms an intermolecular hydrogen bond in the complex interface (shown as a dotted line). (*d*) Hydrophobic interactions of DVL1 Pro657, Pro658 and Gly659. (*e*) DVL1 Pro661 mediates proline–aromatic interactions in the complex interface.

A parallel study performed by Zhao *et al.* has confirmed the binding of KLHL12 to the dopamine D4 receptor and DVL3 by NMR and reported a 2.9 Å crystal structure of the KLHL12 Kelch domain bound to a DVL3 peptide (PDB 6V7O) [36]. Of note, their structure revealed a distinct peptide-binding conformation that probably reflects the fact that the crystallized DVL3 peptide was N-terminally truncated, resulting in the loss of the first proline within the 'PPGAPP' degron sequence (electronic supplementary material, figure S1). The omitted proline position is substituted by a glycine or alanine in partners of KLHL12 outside the DVL1-3 family, which are not targeted for degradation. Thus, it is possible that the structure of Zhao *et al.* captures an alternative peptide-binding mode exploitable by this subgroup of binding partners (electronic supplementary material. figure S2).

Overall, these results define the molecular mechanisms determining DVL regulation by KLHL12 and establish the KLHL12 Kelch domain as a new protein interaction module for proline-rich domains. The BTB-Kelch family E3 ligases also form tractable targets for the design of small molecule inhibitors, or degraders such as PROTACs, which recruit neo-substrates to an E3 ligase for targeted protein degradation

[37]. The structures, identified peptides and functional assays described here represent an important first step in enabling such ligand discovery for KLHL12.

# 4. Methods

## 4.1. Protein expression and purification

Human KLHL12 Kelch domain (Uniprot Q53G59-1, residues 268–567) and KLHL7 Kelch domain (Uniprot Q8IXQ5-1, residues 283–586) were expressed from the vector pNIC28-Bsa4 in BL21(DE3)-R3-pRARE cells and purified by nickel affinity and size exclusion chromatography as described previously [18]. Both plasmids are deposited in addgene (RRID: Addgene_38908 and RRID:Addgene_39023, respectively).

## 4.2. SPOT peptide arrays

Peptide arrays were prepared as described previously [38]. For the initial DVL1 array (figure 1*a*), 20-mer peptides were synthesized directly on a modified cellulose membrane with a

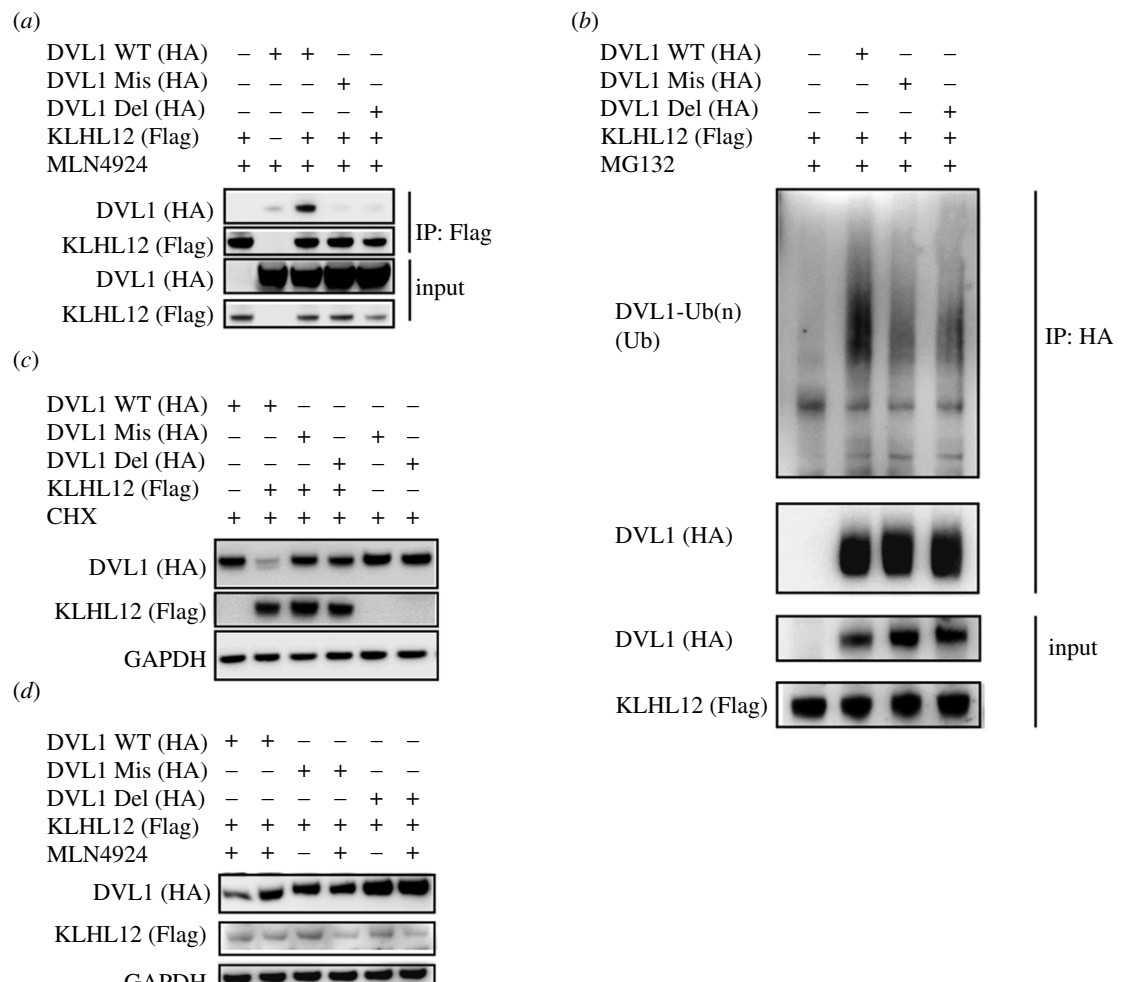

**Figure 5.** Mutations in the DVL1 'PGGPP' motif impair DVL1 binding, ubiquitination and degradation by KLHL12. Two variants of full-length HA-tagged DVL1 were prepared in which the 657-'PPGGPP' motif was either mutated to 'AAAAAA' (DVL1 Mis) or deleted (DVL1 Del). (a) Full-length DVL1 variants and full-length KLHL12 were co-transfected into HEK293T cells treated with MLN4924 as indicated. Flag-KLHL12 immunoprecipitated with anti-Flag antibody bound robustly to DVL1 WT, but not to DVL1 mutants. (b) Mutation or deletion of the 'PGGPP' motif caused a marked reduction in the level of DVL1 ubiquitination. DVL1 variants and full-length KLHL12 were co-transfected into HEK293 cells. Cells were treated with MG132 for 4 h before harvesting to enrich for ubiquitinated forms of DVL1. HA-DVL1 variants were immunoprecipitated with anti-HA agarose gel and poly-ubiquitinated DVL1 detected by anti-ubiquitin (Ub) antibody. (c) The 'PGGPP' sequence regulates DVL1 stability. DVL1 variants were transfected into HEK293T cells with or without full-length KLHL12 as indicated. Cells were treated with cycloheximide (CHX) for 1 hour before harvesting. DVL1 protein levels were detected by western blot and compared to a GAPDH loading control. (d) DVL1 variants and full-length KLHL12 were co-transfected into HEK293T cells. Cells were treated with MLN4924 as indicated for 4 h before harvesting. MLN4924 stabilized DVL1 WT, but not DVL1 mutants.

polyethylglycol linker using the peptide synthesizer MultiPep (Intavis Bioanalytical Instruments AG). Poly-His (20-mer) was used as a positive control at two spots. The membrane was rinsed with ethanol briefly and washed with PBST (1xPBS + 0.1% Tween 20) 3 times for 5 min (3 × 5 min). The membrane was blocked for 1 hour with 5% milk powder in PBST at room temperature, before washing with PBST (2 × 5 min) and PBS (1 × 5 min). The array was then incubated with 0.4 µM His$_6$-tagged Kelch domain protein in PBS for 1 h at 4°C. Unbound protein was washed off in PBST (3 × 5 min) and bound protein was detected using HRP-conjugated anti-His antibody (1 : 2000 in 5% milk PBST; Novagen #71841). After 1 h incubation at room temperature, the membrane was washed (3 × 15 min) using PBST, and ECL kit (Perkin elmer #NEL104) was used before exposing the film for various durations (1, 2, 3, 5, 10 min) and visualized.

Follow-up peptide arrays (figures 2 and 6a) were prepared as described previously [33]. After array synthesis, membranes were incubated with 5% BSA to block nonspecific binding. The arrays were then incubated with 1 µM His$_6$-tagged Kelch domains in PBS at 4°C overnight. Unbound protein was washed off in PBST buffer and bound protein was detected using HRP-conjugated anti-His antibody (Merck Millipore #71840; RRID:AB_1094755).

## 4.3. Fluorescence polarization

Fluorescence polarization assays were performed in 384-well plates using a Synergy 2 microplate reader (BioTek) with excitation and emission wavelengths of 485 nm and 528 nm, respectively. Peptides were synthesized, N-terminally labelled with FITC and purified by Tufts University Core Services (Boston, MA). Binding assays were performed in 10 µl volume at a constant labelled peptide concentration of 40 nM and indicated Kelch domain protein concentrations in a buffer containing 50 mM HEPES, pH 7.5, 300 mM NaCl, 0.5 mM TCEP and 0.01% Triton-X. Binding data were corrected for background of the free labelled peptides (no protein). For $K_D$ determination, data were fitted to a

royalsocietypublishing.org/journal/rsob Open Biol. 10: 200041

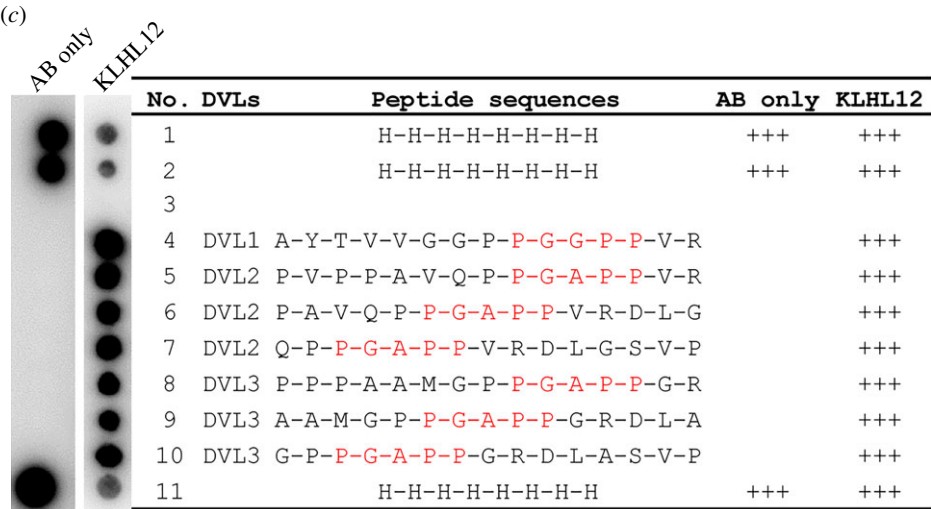

(a)

| | | |
|---|---|---|
| DVL1 | 658- | PGGPP |
| DVL2 | 699- | PGAPP |
| DVL3 | 656- | PGVPP |
| DVL3 | 679- | PGAPP |
| PEF1 | 18- | PGAPP |
| PLEKHA4 | 177- | PGGPP |
| SEC31A | 958- | PGAPP |
| DRD4 | 299- | PGLPP |
| Consensus: | | PGXPP |

(b) WT — G660A —

G660

(c)

AB only   KLHL12

| No. | DVLs | Peptide sequences | AB only | KLHL12 |
|---|---|---|---|---|
| 1 | | H-H-H-H-H-H-H-H | +++ | +++ |
| 2 | | H-H-H-H-H-H-H-H | +++ | +++ |
| 3 | | | | |
| 4 | DVL1 | A-Y-T-V-V-G-G-P-P-G-G-P-P-V-R | | +++ |
| 5 | DVL2 | P-V-P-P-A-V-Q-P-P-G-A-P-P-V-R | | +++ |
| 6 | DVL2 | P-A-V-Q-P-P-G-A-P-P-V-R-D-L-G | | +++ |
| 7 | DVL2 | Q-P-P-G-A-P-P-V-R-D-L-G-S-V-P | | +++ |
| 8 | DVL3 | P-P-P-A-A-M-G-P-P-G-A-P-P-G-R | | +++ |
| 9 | DVL3 | A-A-M-G-P-P-G-A-P-P-G-R-D-L-A | | +++ |
| 10 | DVL3 | G-P-P-G-A-P-P-G-R-D-L-A-S-V-P | | +++ |
| 11 | | H-H-H-H-H-H-H-H | +++ | +++ |

**Figure 6.** 'PGXPP' is a consensus motif in DVL1-3 for KLHL12 interaction. (a) A 'PGXPP' motif is conserved across known KLHL12 binders, including the three DVL paralogs. 'X' refers to a small non-polar amino acid. (b) Structural model incorporating the G660A substitution (yellow) in the DVL1 structure (pink sticks). (c) DVL1, DVL2 and DVL3 peptides containing the 'PGXPP' motif were printed in SPOT peptide arrays. Arrays were incubated with purified 6 x His-KLHL12 Kelch domain, washed and then binding detected using anti-His antibody.

hyperbolic function using Sigma Plot software (Systat Software, San Jose CA). $K_D$ values represent mean ± S.E. ($n = 3$).

## 4.4. Structure determination

KLHL12 Kelch domain was concentrated to 8 mg ml$^{-1}$ in 50 mM HEPES pH 7.5, 300 mM NaCl and 0.5 mM TCEP buffer. DVL1 peptide was added in the same buffer to a final concentration of 3 mM. The protein-peptide mixture was incubated on ice for 1 h prior to setting up sitting-drop vapour-diffusion crystallization plates. Micro-seed stocks were prepared from small KLHL12 crystals grown during previous rounds of crystal optimization. The best-diffracting crystals of the KLHL12 complex were obtained at 4°C by mixing 100 nl protein, 20 nl diluted seed stock and 50 nl of a reservoir solution containing 30% PEG4000, 0.2 M ammonium acetate and 0.1 M acetate pH 4.5. Prior to vitrification in liquid nitrogen, crystals were cryoprotected by direct addition of reservoir solution supplemented with 25% ethylene glycol. Diffraction data were collected at the Diamond Light Source beamline I24. Diffraction data were indexed and integrated using MOSFLM [39], and scaled and merged using AIMLESS [40]. Molecular replacement was performed with Phaser MR [41] in Phenix using KLHL12 apo structure (PDB: 2VPJ) as the model. COOT was used for DVL1 building and manual refinement [42], whereas PHENIX.REFINE was used for automated refinement [43]. Refined models were validated by MolProbity [44]. Structure figures were made with PyMOL [45]. The interaction interfaces were analysed using 'Protein interfaces, surfaces and assemblies' service PISA at the European Bioinformatics Institute [46].

## 4.5. Immunoprecipitation

Full-length DVL1 variants (Uniprot O14640-1) and KLHL12 were subcloned into a pcDNA3 vector, with N-terminal HA tag and Flag tag, respectively. HEK293T cells (ATCC #CRL-3216; RRID:CVCL_0063) were cultured in high-glucose Dulbecco's Modified Eagle's Medium (Sigma-Aldrich) with 5% penicillin streptomycin (ThermoFisher) and 10% Fetal Bovine Serum (Sigma-Aldrich) inside a 5% CO$_2$ incubator at 37°C. Full-length KLHL12 and full-length DVL1 constructs were transfected into HEK293T cells at 60% confluency using polyethylenimine. 40 h after transfection, cells were treated with MLN4924 for four hours and then harvested and lysed in the presence of protease and phosphatase inhibitors. Immunoprecipitation was performed using ANTI-FLAG M2 Affinity Gel (Sigma-Aldrich). Results were analysed using western blotting (Anti-Flag antibody, Sigma-Aldrich, F1804; RRID: AB_262044; Anti-HA antibody, Biolegend, 901501; RRID: AB_2565006).

## 4.6. Ubiquitination assay

Full-length Flag-KLHL12 and HA-DVL1 constructs were transfected into HEK293T cells at 60% confluency with polyethylenimine. 40 h after transfection, cells were treated

with MG132 for 4 h and then harvested and lysed in the presence of 2% SDS. The lysate was sonicated and diluted in buffer containing 1% Triton as described in [47]. DVL1 was immunoprecipitated using Pierce Anti-HA Agarose (Thermo Scientific, 26181). Results were analysed using western blotting (Anti-Flag antibody, Sigma-Aldrich, F1804; Anti-HA antibody, Biolegend, 901501; Anti-Ub antibody, Novus Biologicals, NB300-130).

## 4.7. Stability assays

Full-length Flag-KLHL12 and HA-DVL1 constructs were transfected into HEK293T cells at 60% confluency with polyethylenimine. 40 h after transfection, cells were treated with cycloheximide (CHX) for 1 h or MLN4924 for 4 h and then harvested and lysed. Protein levels were analysed by Western blotting (anti-Flag antibody, Sigma-Aldrich, F1804; RRID: AB_262044 and anti-HA antibody, Biolegend, 901501; RRID: AB_2565006).

## Abbreviations

The abbreviations used are Del, deletion mutant; DVL, dishevelled; FITC, fluorescein isothiocyanate; GST, glutathione S-transferase; HRP, horseradish peroxidase; Mis, missense mutant; N.B, no binding; PBS, phosphate-buffered saline; RMS, root-mean-square; TCEP, tris(2-carboxyethyl)phosphine; Ub, ubiquitin; WT, wild type.

Data accessibility. The coordinates and structure factors for the crystal structure reported in this article have been deposited in the PDB with accession code 6TTK.

Authors' contributions. A.N.B., M.V. and V.D. designed and supervised the research. G.A.W., S.P., P.F. and M.V. performed the peptide arrays. G.A.W. performed the fluorescence polarization experiments. Z.C. performed the cellular experiments and solved the crystal structure. Z.C., M.V. and A.N.B. wrote the paper.

Competing interests. We declare we have no competing interests

Funding. The SGC is a registered charity (number 1097737) that receives funds from AbbVie, Bayer Pharma AG, Boehringer Ingelheim, Canada Foundation for Innovation, Eshelman Institute for Innovation, Genome Canada, Innovative Medicines Initiative (EU/EFPIA) (ULTRA-DD grant no. 115766), Janssen, Merck KGaA Darmstadt Germany, MSD, Novartis Pharma AG, Ontario Ministry of Economic Development and Innovation, Pfizer, São Paulo Research Foundation-FAPESP, Takeda and Wellcome (106169/ZZ14/Z).

Acknowledgements. The authors would like to thank Diamond Light Source for beamtime (proposal mx15433), as well as the staff of beamline I02, I04 and I24 for assistance with crystal testing and data collection.

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
