## [Reviewer comments · Open Biology]

Review History

RSOB-20-0041.R0 (Original submission)

Review form: Reviewer 1

Recommendation

Major revision is needed (please make suggestions in comments)

Do you have any ethical concerns with this paper?

No

Comments to the Author

This manuscript defined the critical region in DVL1 C-terminus that binds to KLHL12 using SPOT peptide technology. Based on this information, they solved the crystal structure of KLHL12 kelch domain in complex with a DVL1 peptide. Data from both peptide binding and structural analyses revealed the importance of PGXPP motif in KLHL12 binding. They then showed the presence of this motif in other KLHL12 substrates and binding partners.

Overall, this is a solid study. Although the crystal structure of KLHL12 kelch domain was reported previously, unraveling the structural insights for the substrate binding mode would

benefit future discovery of small molecular inhibitors of this E3 ligase. The identification of a potential “common degron” motif for KLHL12 substrates/partners is also important for the discovery of new substrates/regulators of this E3 ligase. The major limitation, however, is that a similar study was recently published (Zhao et al., *Biochemistry* 59: 964-969, 2020). This paper reported the structure of KLHL12 kelch domain in complex with a DVL3 peptide and reached to the same finding and conclusion.

If the authors can emphasize/strengthen the “common degron” idea by including peptides from more KLHL12 substrates/partners as well as their mutants (e.g. alanine mutations in the PGXPP motif) in Fig. 6C, this study would provide information that was not covered in the paper by Zhao et al.

Review form: Reviewer 2

Recommendation

Accept with minor revision (please list in comments)

Do you have any ethical concerns with this paper?

No

Comments to the Author

Dishevelled proteins (Dvl1-3) play a crucial role in Wnt signaling that regulates the early development and growth of many human cancers. So, it is vital to understand how binding partners regulate Dvl's proteins. Recent studies showed that Dvl1 proteins could interact with KLHL12 proteins; the binding mechanism was not precise. Now, the authors presented the X-ray crystal structure of the Kelch domain of KLHL12 in complex with a Dvl1 peptide.

To elucidate the interaction, the authors used the SPOT peptide technology, fluorescence polarization assay, and cell assay, and X-ray crystallography, showing that the binding PGXPP motif in Dvl1 protein is critical to the binding of the Kelch domain of KLHL12.

Very recently, Bin Zhao et al. from Vanderbilt University, USA (*Biochemistry* 2020, 59, 8, 964-969) also reported the X-ray structure of the Kelch domain of KLHL-12 in complex with a Dvl 3 peptide. Their conclusion in *biochemistry* was similar to this manuscript, although the authors by this manuscript provided more extensive and concrete data.

So, I would like to recommend for the authors to discuss Zhao et al's work in a modified version of the manuscript.

Review form: Reviewer 3

Recommendation

Accept with minor revision (please list in comments)

Do you have any ethical concerns with this paper?

No

Comments to the Author

This is a very nice paper describing work well-done to study the E3 ligase-substrate degron interaction for the Cul3 ligase KLHL12 and substrate Disheveled, which plays a role in the Wnt/beta catenin signaling. The authors mapped the degron sequence on substrate DVL1 using

SPOT peptide array and elegant Ala scan and truncation experiments. They then co-crystallized the Kelch domain of KLHL12 with a 15-mer substrate peptide spanning the identified PPGGP motif bound ($K_d \sim 20 \mu\text{M}$). The high res co-structure confirmed the expected binding site on the Kelch domain and clarified the detailed interactions at atomic level. They then go on to perform biochemical pull down and ubiquitination experiments in cells to confirm the physiological relevance of the degron-ligase interaction

The study is well-done and the conclusions are supported by the shown data. The knowledge of the revealed “degron” binding mode is important because it provides the molecular basis for ligase-substrate recognition on a biologically important Cul3 system, and a starting point for ligand design. There are also interesting features such as the tight turn and SAR that are revealed.

Overall, we recommend publication in the journal, and congratulate the authors on this nice piece of work. Prior to publication, we would also like to offer an opportunity to improve the manuscript with a revision based on the comments below.

Major comments:

a) The main point of attention is that a related structure paper was recently published by the Fesik group (Zhao et al. *Biochemistry* 2020; DOI: 10.1021/acs.biochem.9b01073). While that paper might be perceived to somewhat detract from the novelty of the current paper under review, the findings reported by Bullock et al. certainly warrant publication of their paper. Such competitive situations occur not infrequently, particularly in the field of E3 ligases these days, and the authors submitting around the same time as the others should be given a fair chance to report their work. This work was carried out independently and simultaneously after all. Nevertheless, we feel it important that the authors acknowledge this article, and the community will appreciate that this group fairly references the work of others. The authors might also want to compare and contrast the findings. It turns out, the higher resolution of their new structure in comparison with that of the Zhao KLHL12 paper, and structural superposition might reveal something interesting.

b) As the authors are well aware, there is growing interest in targeting E3 ligases with inhibitors and degraders e.g. PROTACs, so enabling new E3 ligases by “de-orphanising” them with new non-covalent ligands such as degron peptides is an important first direction. This is clearly an important first step enabling ligand discovery on this target. The authors might want to mention this point as a outlook to the future.

Minor comments:

1. in the abstract, presumably mutation/deletion of the motif should cause an increase (not a reduction) in stability of DVL1
2. pg.4, The word "finally" in the sentence “Transfer of ubiquitin from the E2...” creates ambiguity. the authors might want to remove this word
3. pg.5 the authors might want to cite Ronald Frank SPOT assay synthesis paper
4. last sentence of pg. 5: Could the authors comment on the apparently stronger binding to the truncated peptides in Figure 2B? In particular, truncating to Pro657 and the N-terminus and Arg664 at the C-terminus. As noted by Zhao et al, this could be due to a conformational entropy penalty with longer peptides
5. pg. 6, last paragraph of section “Structure determination..”: Pro662 (PGXPP) is not built in chain F due to a lack of density and the density for this residue is poor in all the other chains. It also doesn't appear to interact with KLHL12 when it is resolved while the Pro preceding their proposed PGXPP motif is clearly resolved. This is alluded to in the next section but perhaps the

lack of density could be mentioned as it suggests flexibility at this residue.

6. pg. 7: "partially positive π face of the pyrrolidine ring" is incorrect, because the pyrrolidine ring of proline is not aromatic so does not have a π face. For an accurate account of Aromatic-Proline Interactions see <https://doi.org/10.1021/ar300087y>

7. pg. 8 and Figure 5B: There seems to be still some background ubiquitination with the DVL1 mutants. The authors might want to comment on why that might be. Which other ligases are likely responsible or is it endogenous KLHL12? It could be clearer if experiments could be performed in KLHL12 knockout cells, assuming knockouts are viable. Also, controls that include MLN4924 and MG132 would be useful. If these experiments cannot be performed, the authors might want to comment and acknowledge it as a caveat / limitation

8. The methods for the experiments described in Figure 5C-D seem to be written in the figure legends but not detailed in methods section.

9. end of Figure 1 legend: could specify here that the error bars are standard error as mentioned in the methods

10. Table 1: Data collection and refinement statistics all look fine, but there are several sodium atoms and a single chlorine atom built in the the structure. How convinced are the authors that these aren't just waters?

We agree to waive our anonymity,
Alessio Ciulli and Angus Cowan

Decision letter (RSOB-20-0041.R0)

01-May-2020

Dear Dr Bullock,

We are pleased to inform you that your manuscript RSOB-20-0041 entitled "Identification of a PGXPP degron motif in dishevelled and structural basis for its binding to the E3 ligase KLHL12" has been accepted by the Editor for publication in Open Biology. The reviewer(s) have recommended publication, but also suggest some minor revisions to your manuscript. Therefore, we invite you to respond to the reviewer(s)' comments and revise your manuscript.

Please submit the revised version of your manuscript within 7 days. If you do not think you will be able to meet this date please let us know immediately and we can extend this deadline for you.

When submitting your revised manuscript, you will be able to respond to the comments made by the referee(s) and upload a file "Response to Referees" in "Section 6 - File Upload". You can use this to document any changes you make to the original manuscript. In order to expedite the

processing of the revised manuscript, please be as specific as possible in your response to the referee(s).

- 1) A text file of the manuscript (doc, txt, rtf or tex), including the references, tables (including captions) and figure captions. Please remove any tracked changes from the text before submission. PDF files are not an accepted format for the "Main Document".
- 2) A separate electronic file of each figure (tiff, EPS or print-quality PDF preferred). The format should be produced directly from original creation package, or original software format. Please note that PowerPoint files are not accepted.
- 3) Electronic supplementary material: this should be contained in a separate file from the main text and meet our ESM criteria (see <http://royalsocietypublishing.org/instructions-authors#question5>). All supplementary materials accompanying an accepted article will be treated as in their final form. They will be published alongside the paper on the journal website and posted on the online figshare repository. Files on figshare will be made available approximately one week before the accompanying article so that the supplementary material can be attributed a unique DOI.

Online supplementary material will also carry the title and description provided during submission, so please ensure these are accurate and informative. Note that the Royal Society will not edit or typeset supplementary material and it will be hosted as provided. Please ensure that the supplementary material includes the paper details (authors, title, journal name, article DOI). Your article DOI will be 10.1098/rsob.2016[*last 4 digits of e.g. 10.1098/rsob.20160049*].

- 4) A media summary: a short non-technical summary (up to 100 words) of the key findings/importance of your manuscript. Please try to write in simple English, avoid jargon, explain the importance of the topic, outline the main implications and describe why this topic is newsworthy.

Images

Data-Sharing

It is a condition of publication that data supporting your paper are made available. Data should be made available either in the electronic supplementary material or through an appropriate repository. Details of how to access data should be included in your paper. Please see <https://royalsocietypublishing.org/rsob/for-authors#question4> for more details.

Data accessibility section

Sincerely,
The Open Biology Team
mailto:openbiology@royalsociety.org

Reviewer(s)' Comments to Author:

Referee: 1

Comments to the Author(s)

This manuscript defined the critical region in DVL1 C-terminus that binds to KLHL12 using SPOT peptide technology. Based on this information, they solved the crystal structure of KLHL12 kelch domain in complex with a DVL1 peptide. Data from both peptide binding and structural analyses revealed the importance of PGXPP motif in KLHL12 binding. They then showed the presence of this motif in other KLHL12 substrates and binding partners.

Overall, this is a solid study. Although the crystal structure of KLHL12 kelch domain was reported previously, unraveling the structural insights for the substrate binding mode would benefit future discovery of small molecular inhibitors of this E3 ligase. The identification of a potential "common degron" motif for KLHL12 substrates/partners is also important for the discovery of new substrates/regulators of this E3 ligase. The major limitation, however, is that a similar study was recently published (Zhao et al., *Biochemistry* 59: 964-969, 2020). This paper reported the structure of KLHL12 kelch domain in complex with a DVL3 peptide and reached to the same finding and conclusion.

If the authors can emphasize/strengthen the "common degron" idea by including peptides from more KLHL12 substrates/partners as well as their mutants (e.g. alanine mutations in the PGXPP motif) in Fig. 6C, this study would provide information that was not covered in the paper by Zhao et al.

Referee: 2

Comments to the Author(s)

Dishevelled proteins (Dvl1-3) play a crucial role in Wnt signaling that regulates the early development and growth of many human cancers. So, it is vital to understand how binding partners regulate Dvl's proteins. Recent studies showed that Dvl1 proteins could interact with KLHL12 proteins; the binding mechanism was not precise. Now, the authors presented the X-ray crystal structure of the Kelch domain of KLHL12 in complex with a Dvl1 peptide.

To elucidate the interaction, the authors used the SPOT peptide technology, fluorescence polarization assay, and cell assay, and X-ray crystallography, showing that the binding PGxPP motif in Dvl1 protein is critical to the binding of the Kelch domain of KLHL12.

Very recently, Bin Zhao et al. from Vanderbilt University, USA (*Biochemistry* 2020, 59, 8, 964-969) also reported the X-ray structure of the Kelch domain of KLHL-12 in complex with a Dvl 3 peptide. Their conclusion in biochemistry was similar to this manuscript, although the authors by this manuscript provided more extensive and concrete data.

So, I would like to recommend for the authors to discuss Zhao et al's work in a modified version of the manuscript.

Referee: 3

Comments to the Author(s)

This is a very nice paper describing work well-done to study the E3 ligase-substrate degron interaction for the Cul3 ligase KLHL12 and substrate Disheveled, which plays a role in the Wnt/beta catenin signaling. The authors mapped the degron sequence on substrate DVL1 using SPOT peptide array and elegant Ala scan and truncation experiments. They then co-crystallized the Kelch domain of KLHL12 with a 15-mer substrate peptide spanning the identified PPGGP motif bound ($K_d \sim 20 \mu\text{M}$). The high res co-structure confirmed the expected binding site on the Kelch domain and clarified the detailed interactions at atomic level. They then go on to perform biochemical pull down and ubiquitination experiments in cells to confirm the physiological relevance of the degron-ligase interaction

The study is well-done and the conclusions are supported by the shown data. The knowledge of the revealed "degron" binding mode is important because it provides the molecular basis for ligase-substrate recognition on a biologically important Cul3 system, and a starting point for ligand design. There are also interesting features such as the tight turn and SAR that are revealed.

Overall, we recommend publication in the journal, and congratulate the authors on this nice piece of work. Prior to publication, we would also like to offer an opportunity to improve the manuscript with a revision based on the comments below.

Major comments:

a) The main point of attention is that a related structure paper was recently published by the Fesik group (Zhao et al. *Biochemistry* 2020; DOI: 10.1021/acs.biochem.9b01073). While that paper might be perceived to somewhat detract from the novelty of the current paper under review, the findings reported by Bullock et al. certainly warrant publication of their paper. Such competitive situations occur not infrequently, particularly in the field of E3 ligases these days, and the authors submitting around the same time as the others should be given a fair chance to report their work. This work was carried out independently and simultaneously after all. Nevertheless, we feel it important that the authors acknowledge this article, and the community will appreciate that this group fairly references the work of others. The authors might also want to compare and contrast the findings. It turns out, the higher resolution of their new structure in comparison with that of the Zhao KLHL12 paper, and structural superposition might reveal something interesting.

b) As the authors are well aware, there is growing interest in targeting E3 ligases with inhibitors and degraders e.g. PROTACs, so enabling new E3 ligases by "de-orphanising" them with new non-covalent ligands such as degron peptides is an important first direction. This is clearly an important first step enabling ligand discovery on this target. The authors might want to mention this point as an outlook to the future.

Minor comments:

1. in the abstract, presumably mutation/deletion of the motif should cause an increase (not a reduction) in stability of DVL1

2. pg.4, The word "finally" in the sentence "Transfer of ubiquitin from the E2..." creates ambiguity. the authors might want to remove this word

3. pg.5 the authors might want to cite Ronald Frank SPOT assay synthesis paper

4. last sentence of pg. 5: Could the authors comment on the apparently stronger binding to the truncated peptides in Figure 2B? In particular, truncating to Pro657 and the N-terminus and

Arg664 at the C-terminus. As noted by Zhao et al, this could be due to a conformational entropy penalty with longer peptides

5. pg. 6, last paragraph of section "Structure determination..": Pro662 (PGXPP) is not built in chain F due to a lack of density and the density for this residue is poor in all the other chains. It also doesn't appear to interact with KLHL12 when it is resolved while the Pro preceding their proposed PGXPP motif is clearly resolved. This is alluded to in the next section but perhaps the lack of density could be mentioned as it suggests flexibility at this residue.

6. pg. 7: "partially positive p face of the pyrrolidine ring" is incorrect, because the pyrrolidine ring of proline is not aromatic so does not have a p face. For an accurate account of Aromatic-Proline Interactions see <https://doi.org/10.1021/ar300087y>

7. pg. 8 and Figure 5B: There seems to be still some background ubiquitination with the DVL1 mutants. The authors might want to comment on why that might be. Which other ligases are likely responsible or is it endogenous KLHL12? It could be clearer if experiments could be performed in KLHL12 knockout cells, assuming knockouts are viable. Also, controls that include MLN4924 and MG132 would be useful. If these experiments cannot be performed, the authors might want to comment and acknowledge it as a caveat / limitation

8. The methods for the experiments described in Figure 5C-D seem to be written in the figure legends but not detailed in methods section.

9. end of Figure 1 legend: could specify here that the error bars are standard error as mentioned in the methods

10. Table 1: Data collection and refinement statistics all look fine, but there are several sodium atoms and a single chlorine atom built in the the structure. How convinced are the authors that these aren't just waters?

We agree to waive our anonymity,
Alessio Ciulli and Angus Cowan

Author's Response to Decision Letter for (RSOB-20-0041.R0)

See Appendix A.

Decision letter (RSOB-20-0041.R1)

14-May-2020

Dear Dr Bullock,

We are pleased to inform you that your manuscript entitled "Identification of a PGXPP degron motif in dishevelled and structural basis for its binding to the E3 ligase KLHL12" has been accepted by the Editor for publication in Open Biology.

Article processing charge

Please note that the article processing charge is immediately payable. A separate email will be sent out shortly to confirm the charge due. The preferred payment method is by credit card; however, other payment options are available.

Sincerely,

The Open Biology Team
mailto: openbiology@royalsociety.org

Appendix A

RSOB-20-0041: Identification of a PGXPP degron motif in dishevelled and structural basis for its binding to the E3 ligase KLHL12

Response to reviewers:

Referee: 1

Comments to the Author(s)

This manuscript defined the critical region in DVL1 C-terminus that binds to KLHL12 using SPOT peptide technology. Based on this information, they solved the crystal structure of KLHL12 kelch domain in complex with a DVL1 peptide. Data from both peptide binding and structural analyses revealed the importance of PGXPP motif in KLHL12 binding. They then showed the presence of this motif in other KLHL12 substrates and binding partners.

Overall, this is a solid study. Although the crystal structure of KLHL12 kelch domain was reported previously, unraveling the structural insights for the substrate binding mode would benefit future discovery of small molecular inhibitors of this E3 ligase. The identification of a potential “common degron” motif for KLHL12 substrates/partners is also important for the discovery of new substrates/regulators of this E3 ligase. The major limitation, however, is that a similar study was recently published (Zhao et al., *Biochemistry* 59: 964-969, 2020). This paper reported the structure of KLHL12 kelch domain in complex with a DVL3 peptide and reached to the same finding and conclusion.

If the authors can emphasize/strengthen the “common degron” idea by including peptides from more KLHL12 substrates/partners as well as their mutants (e.g. alanine mutations in the PGXPP motif) in Fig. 6C, this study would provide information that was not covered in the paper by Zhao et al.

We thank the reviewer for their comments. We have prepared two new figures (Figure S1 and Figure S2) that explore the common degron through a detailed comparison of our work with that of Zhao et al. We note that the respective crystal structures reveal two distinct peptide-binding modes. All the DVL1-3 degron motifs contain four prolines within a core PPGXPP motif. Our structure using a 15-mer DVL1 peptide reveals buried interface positions for the first, second and third prolines in this motif, while the fourth proline is largely exposed. By comparison, the crystallised DVL3 peptide used by Zhao et al was truncated at the N-terminus and omitted the first proline. This truncation appears to have induced a shift in the bound peptide position such that the conserved motif's second, third and fourth prolines are now buried. We note that the other partners and substrates of KLHL12 have a comparable XPGXPP motif, where the first proline is substituted to Gly or Ala (represented by X). We speculate that our 2.4 Å structure captures the true representation of the DVL substrate interactions, whereas the other partners have potential to adopt either binding mode, likely depending on their N-terminal sequences. The N-terminal truncation of the DVL3 peptide of Zhao et al also prevents the formation of two intramolecular hydrogen bond interactions that stabilise the bound DVL1 conformation in our own structure. This likely introduces further subtle conformational differences between the two peptide structures.

Referee: 2

Comments to the Author(s)

Dishevelled proteins (Dvl1-3) play a crucial role in Wnt signaling that regulates the early development and growth of many human cancers. So, it is vital to understand how binding partners regulate Dvl's proteins. Recent studies showed that Dvl1 proteins could interact with KLHL12 proteins; the binding mechanism was not

precise. Now, the authors presented the X-ray crystal structure of the Kelch domain of KLHL12 in complex with a Dvl1 peptide.

To elucidate the interaction, the authors used the SPOT peptide technology, fluorescence polarization assay, and cell assay, and X-ray crystallography, showing that the binding PGxPP motif in Dvl1 protein is critical to the binding of the Kelch domain of KLHL12.

Very recently, Bin Zhao et al. from Vanderbilt University, USA (Biochemistry 2020, 59, 8, 964-969) also reported the X-ray structure of the Kelch domain of KLHL-12 in complex with a Dvl 3 peptide. Their conclusion in biochemistry was similar to this manuscript, although the authors by this manuscript provided more extensive and concrete data.

So, I would like to recommend for the authors to discuss Zhao et al's work in a modified version of the manuscript.

We thank the reviewer for their comments. In our revised manuscript, we have discussed Zhao et al's work and made detailed comparisons in new Figures S1 and S2 to address some differences and their implications, as outlined above to reviewer #1.

Referee: 3

Comments to the Author(s)

This is a very nice paper describing work well-done to study the E3 ligase-substrate degron interaction for the Cul3 ligase KLHL12 and substrate Disheveled, which plays a role in the Wnt/beta catenin signaling. The authors mapped the degron sequence on substrate DVL1 using SPOT peptide array and elegant Ala scan and truncation experiments. They then co-crystallized the Kelch domain of KLHL12 with a 15-mer substrate peptide spanning the identified PPGGP motif bound ($K_d \sim 20\mu M$). The high res co-structure confirmed the expected binding site on the Kelch domain and clarified the detailed interactions at atomic level. They then go on to perform biochemical pull down and ubiquitination experiments in cells to confirm the physiological relevance of the degron-ligase interaction

The study is well-done and the conclusions are supported by the shown data. The knowledge of the revealed "degron" binding mode is important because it provides the molecular basis for ligase-substrate recognition on a biologically important Cul3 system, and a starting point for ligand design. There are also interesting features such as the tight turn and SAR that are revealed.

Overall, we recommend publication in the journal, and congratulate the authors on this nice piece of work. Prior to publication, we would also like to offer an opportunity to improve the manuscript with a revision based on the comments below.

Major comments:

a) The main point of attention is that a related structure paper was recently published by the Fesik group (Zhao et al. Biochemistry 2020; DOI: 10.1021/acs.biochem.9b01073). While that paper might be perceived to somewhat detract from the novelty of the current paper under review, the findings reported by Bullock et al. certainly warrant publication of their paper. Such competitive situations occur not infrequently, particularly in the field of E3 ligases these days, and the authors submitting around the same time as the others should be given a fair chance

to report their work. This work was carried out independently and simultaneously after all. Nevertheless, we feel it important that the authors acknowledge this article, and the community will appreciate that this group fairly references the work of others. The authors might also want to compare and contrast the findings. It turns out, the higher resolution of their new structure in comparison with that of the Zhao KLHL12 paper, and structural superposition might reveal something interesting.

We thank the reviewers for their comments. In the revised manuscript, we have compared Zhao et al's structure and ours in new Figure S1 and addressed the implications of the potential plasticity of the KLHL12 interaction in new Figure S2. As outlined to reviewer #1 above, we speculate that the truncated peptide used by Zhao et al may reveal an alternative degron binding mode, potentially more suited to KLHL12-binding partners outside the DVL1-3 family.

b) As the authors are well aware, there is growing interest in targeting E3 ligases with inhibitors and degraders e.g. PROTACs, so enabling new E3 ligases by “de-orphanising” them with new non-covalent ligands such as degron peptides is an important first direction. This is clearly an important first step enabling ligand discovery on this target. The authors might want to mention this point as a outlook to the future.

Following the reviewers' suggestion we have added the following text to the end of the discussion *'The BTB-Kelch family E3 ligases also form tractable targets for the design of small molecule inhibitors, or degraders such as PROTACs, which recruit neo-substrates to an E3 ligase for targeted protein degradation [36]. The structures, identified peptides and functional assays described here represent an important first step in enabling such ligand discovery for KLHL12.'*

Minor comments:

1. in the abstract, presumably mutation/deletion of the motif should cause an increase (not a reduction) in stability of DVL1

We thank the reviewers for spotting this error. We have corrected the text to *'In cells, the mutation or deletion of this motif reduced the binding and ubiquitination of DVL1 and increased its stability confirming this sequence as a degron motif for KLHL12 recruitment'*. We have similarly corrected this point in the discussion section.

2. pg.4, The word "finally" in the sentence “Transfer of ubiquitin from the E2....” creates ambiguity. the authors might want to remove this word

We have deleted the word 'finally' as suggested.

3. pg.5 the authors might want to cite Ronald Frank SPOT assay synthesis paper

We have cited Ronald Frank's SPOT assay paper (Tetrahedron 1992) as new ref 29.

4. last sentence of pg. 5: Could the authors comment on the apparently stronger binding to the truncated peptides in Figure 2B? In particular, truncating to Pro657 and the N-terminus and Arg664 at the C-terminus. As noted by Zhao et al, this could be due to a conformational entropy penalty with longer peptides

We have added the following comment to the Figure 2B legend *'The apparent stronger binding of some truncated peptides may reflect a conformational entropic*

penalty for longer peptides, or differences in the efficiency of peptide synthesis or their accessibility on the membrane’.

5. pg. 6, last paragraph of section “Structure determination..”: Pro662 (PGXPP) is not built in chain F due to a lack of density and the density for this residue is poor in all the other chains. It also doesn’t appear to interact with KLHL12 when it is resolved while the Pro preceding their proposed PGXPP motif is clearly resolved. This is alluded to in the next section but perhaps the lack of density could be mentioned as it suggests flexibility at this residue.

We have added the comment as suggested with the new text ‘Electron density for Pro662 in chains E, G and H was not as clearly resolved as that of the preceding prolines, suggesting a flexible peptide conformation at this position.’

6. pg. 7: “partially positive π face of the pyrrolidine ring” is incorrect, because the pyrrolidine ring of proline is not aromatic so does not have a π face. For an accurate account of Aromatic–Proline Interactions see <https://doi.org/10.1021/ar300087y>

We have removed this erroneous text as suggested.

7. pg. 8 and Figure 5B: There seems to be still some background ubiquitination with the DVL1 mutants. The authors might want to comment on why that might be. Which other ligases are likely responsible or is it endogenous KLHL12? It could be clearer if experiments could be performed in KLHL12 knockout cells, assuming knockouts are viable. Also, controls that include MLN4924 and MG132 would be useful. If these experiments cannot be performed, the authors might want to comment and acknowledge it as a caveat / limitation

We have added the comment ‘Some remaining background ubiquitination of the two DVL1 mutants may result from other endogenous E3 ligases, such as ITCH, NEDDL4 and VHL [9-11], that likely recognise other distinct degron motifs in DVL1.’

8. The methods for the experiments described in Figure 5C-D seem to be written in the figure legends but not detailed in methods section.

We have added the new section ‘Stability assays’ in the method section. The new text reads ‘Full length Flag-KLHL12 and HA-DVL1 constructs were transfected into HEK293T cells at 60% confluency with polyethylenimine. 40 hours after transfection, cells were treated with cycloheximide (CHX) for 1 hour or MLN4924 for 4 hours and then harvested and lysed. Protein levels were analysed by Western blotting (Anti-Flag antibody, Sigma-Aldrich, F1804; Anti-HA antibody, Biolegend, 901501).’

9. end of Figure 1 legend: could specify here that the error bars are standard error as mentioned in the methods

Done.

10. Table 1: Data collection and refinement statistics all look fine, but there are several sodium atoms and a single chlorine atom built in the the structure. How convinced are the authors that these aren't just waters?

The solvent ions Na⁺ and Cl⁻ were modelled according to their fitness to observed and calculated electron density, with the surrounding chemistry taken into consideration. Since the solvent ions did not appear to be significant to the protein function, further experiments, such as iodide crystal soaking were not performed.